# Graph In-Context Operator Networks for Generalizable Spatiotemporal Prediction

## Abstract

In-context operator learning enables neural networks to infer solution operators from contextual examples without weight updates. While prior work has demonstrated the effectiveness of this paradigm in leveraging vast datasets, a systematic comparison against single-operator learning using identical training data has been absent. We address this gap through controlled experiments comparing in-context operator learning against classical operator learning (single-operator models trained without contextual examples), under the same training steps and dataset. To enable this investigation on real-world spatiotemporal systems, we propose GICON (Graph In-Context Operator Network), combining graph message passing for geometric generalization with example-aware positional encoding for cardinality generalization. Experiments on air quality prediction across two Chinese regions show that in-context operator learning outperforms classical operator learning on complex tasks, generalizing across spatial domains and scaling robustly from few training examples to 100 at inference.

## 1. Introduction

Deep learning for partial differential equations (PDEs) has progressed from solution approximation methods (E et al., 2017; Sirignano & Spiliopoulos, 2018; Raissi et al., 2019; Long et al., 2018) that require retraining for each instance, through operator learning (Lu et al., 2021; Li et al., 2021) that generalises across input functions for a fixed operator family, to in-context operator learning (Yang et al., 2023) that infers new operators from contextual examples without weight updates. In-context operator networks (ICONs) condition on key–value pairs[1] to perform operator inference in a single forward pass, enabling a single model to handle diverse operator families. Extensions include language-model-style next-function prediction (Yang et al., 2025),

generalisation to unseen PDE forms (Yang & Osher, 2024), probabilistic modelling (Zhang et al., 2025), and patchified representations for dense 2D problems (Cao et al., 2026). Graph neural networks (Pfaff et al., 2021; Li et al., 2023; Mousavi et al., 2025) offer complementary geometric flexibility through message passing on irregular domains but learn fixed operators. A full discussion of related work is provided in Appendix F.

Despite this progress, a systematic comparison of in-context versus classical single-operator learning under identical training data and compute is absent; existing evaluations (Yang & Osher, 2024; Cao et al., 2026) use different training sets for the two paradigms. Moreover, extending in-context operator learning to real-world spatiotemporal systems introduces two practical challenges: (1) vanilla ICONs (Yang et al., 2023) represent functions as point sequences, which is prohibitive for dense spatial data, while VICON (Cao et al., 2026)'s patchified representations assume regular grids and domain geometries—unsuitable for irregularly sampled sensor networks; and (2) existing models employ fixed positional encodings tied to the training example count, preventing cardinality generalisation at inference.

To address these challenges and enable a controlled comparison, we propose GICON (Graph In-Context Operator Network) with two architectural innovations:

1. **Graph message passing for geometric generalisation and scalability**: We replace patchified representations with graph representations and decompose each GICON layer into spatial message passing and per-node cross-example attention, encoding geometric structure while avoiding joint attention over all nodes and examples.

2. **Example-aware positional encoding for cardinality generalisation**: Inter-example distinction via content-derived attention biases and key–value distinction via learnable offsets enable models trained with few examples (0–5) to generalise stably to 100 at inference.

We validate GICON on air quality prediction (Wang et al., 2025a) across two Chinese regions. Key findings:

- **Geometric generalisation**: Models trained on one re-

---

[1] Prior ICON literature (Yang et al., 2023; 2025; Yang & Osher, 2024; Cao et al., 2026) uses "Cond" and "QoI"; we adopt key–value terminology following the Transformer convention.

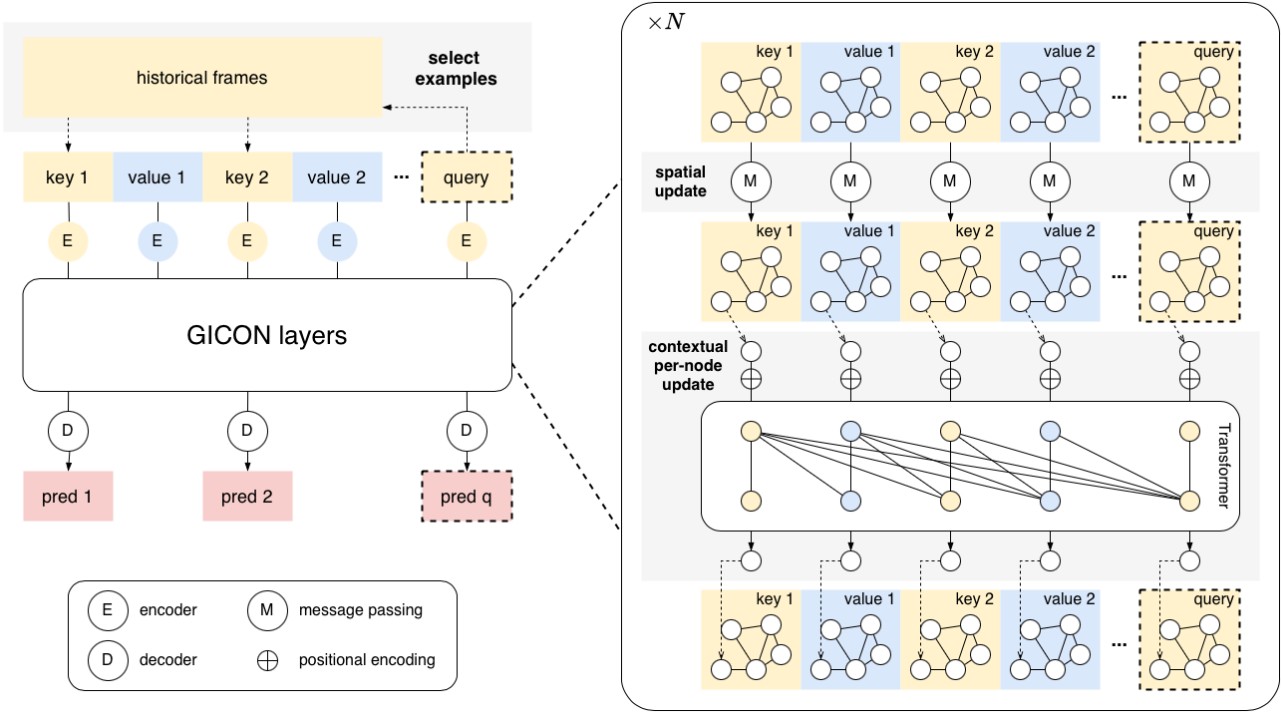

*Figure 1.* GICON architecture. **Left**: Contextual examples form an interleaved key–value sequence processed by separate encoders, $N$ GICON layers, and a decoder. **Right**: Each layer combines spatial message passing (aggregating neighbour information within each graph) with per-node in-context learning (transformer across the example sequence with positional encodings).

gion generalise effectively to another with different graph structures and spatial geometries, without fine-tuning.

- **Cardinality generalisation**: Models trained with at most 5 examples show stable, generally improving performance with up to 100 at inference.

- **In-context operator learning outperforms classical operator learning on complex tasks**: Under the same training steps and dataset, multi-operator in-context learning surpasses single-operator learning on complex operators, with performance improving as example count increases, even for out-of-distribution operators.

## 2. Method

**Problem setup.** We consider spatiotemporal dynamics on $\Omega \subseteq \mathbb{R}^2$ with state $\mathbf{u}(\mathbf{x}, t) \in \mathbb{R}^c$, represented on a graph $\mathcal{G} = (\mathcal{V}, \mathcal{E})$ where nodes correspond to sensor locations and edges encode spatial proximity, naturally accommodating irregular sampling. Since sparse discrete sampling breaks the Markovian property (a single snapshot is insufficient to determine future evolution), we use historical windows $\{\mathbf{u}_{t-\tau+1}, \ldots, \mathbf{u}_t\}$ as inputs. The dynamics are modelled by evolution operators $\{\mathcal{F}_{\Delta t}\}_{\Delta t > 0}$:

$$\mathbf{u}_{t+\Delta t} = \mathcal{F}_{\Delta t}(\mathbf{u}_{t-\tau+1}, \ldots, \mathbf{u}_t). \quad (1)$$

Following the in-context paradigm (Yang et al., 2023), each contextual example is a key–value pair $(\mathbf{k}^{(j)}, \mathbf{v}^{(j)})$ where keys are historical observations and values are future states sharing the same $\Delta t$. The model processes $k$ such examples and a query key to predict the query value in a single forward pass without weight updates. A causal attention mask enables efficient training with varying example counts; all examples within a sequence share the same operator, while operators vary across training sequences.

### 2.1. Retrieval of Examples

To select contextually relevant examples from a large pool, we implement a similarity-based retrieval mechanism. Given a query key and example pool, we extract features by globally averaging the last $\tau_r$ frames over the spatial dimension and compute cosine similarity:

$$s_j = \frac{\mathbf{z}_{\text{query}}^\top \mathbf{z}_{\text{ex}}^{(j)}}{\|\mathbf{z}_{\text{query}}\|_2 \|\mathbf{z}_{\text{ex}}^{(j)}\|_2}. \quad (2)$$

We use FAISS (Douze et al., 2025) for efficient nearest neighbor search. A two-stage selection process first retrieves the top-$K$ most similar examples, then randomly samples $k \leq K$ during training to introduce diversity and prevent overfitting to specific example combinations. Full details of the problem formulation, retrieval mechanism,

and architecture are provided in Appendix A.

## 2.2. Network Architecture

Our GICON integrates graph message passing with in-context operator learning. Given the sampled examples $\mathcal{D}_k = \{(\mathbf{k}^{(j)}, \mathbf{v}^{(j)})\}_{j=1}^k$ and query $\mathbf{k}_{\text{query}}$, we process these as an interleaved sequence to predict $\tilde{\mathbf{v}}_{\text{query}}$.

**Input Projection and Sequence Construction.** Each key (consisting of $\tau$ historical frames) is flattened along the feature dimension and projected via $\text{proj}_{\mathbf{k}} : \mathbb{R}^{\tau \times d_{\text{in}}} \to \mathbb{R}^{d_{\text{node}}}$; each value (a single frame) via $\text{proj}_{\mathbf{v}} : \mathbb{R}^{d_{\text{in}}} \to \mathbb{R}^{d_{\text{node}}}$. The interleaved sequence $[\mathbf{k}^{(1)}, \mathbf{v}^{(1)}, \ldots, \mathbf{k}^{(k)}, \mathbf{v}^{(k)}, \mathbf{k}_{\text{query}}]$ has length $2k + 1$.

**Graph Message Passing with In-Context Learning.** The core of GICON consists of $L$ stacked layers, each combining spatial message passing with in-context learning. Each layer maintains hidden states $\mathbf{h}^{(\ell)} \in \mathbb{R}^{(2k+1) \times |\mathcal{V}| \times d_{\text{node}}}$ and performs two sequential operations:

1. **Spatial Message Passing:** For each sequence position (in parallel), information is aggregated from graph neighbours:

$$\mathbf{m}_{t,i} = \sum_{j \in \mathcal{N}(i)} \text{MLP}_{\text{msg}}([\mathbf{h}_{t,i}^{(\ell)}, \mathbf{h}_{t,j}^{(\ell)}, \mathbf{e}_{ij}]) \quad (3)$$

where $\mathcal{N}(i)$ denotes the neighbours of node $i$ and $\mathbf{e}_{ij}$ is the edge feature. The update is $\tilde{\mathbf{h}}_{t,i}^{(\ell)} = \mathbf{h}_{t,i}^{(\ell)} + \mathbf{m}_{t,i}$.

2. **Per-Node In-Context Learning:** For each node (in parallel), a transformer operates across the sequence dimension with causal masking:

$$\mathbf{h}_i^{(\ell+1)} = \text{Transformer}(\tilde{\mathbf{h}}_i^{(\ell)}, \text{mask}_{\text{causal}}) \quad (4)$$

This design enables each node to learn from spatial neighbours while simultaneously performing in-context learning across examples, combining the geometric flexibility of GNNs with the adaptability of in-context learning.

**Positional Encoding for Example Distinction.** A critical challenge is distinguishing (1) different contextual examples from the query, and (2) keys from values. We introduce two complementary strategies.

For inter-example distinction, we extract key tokens (at even positions) and pool across nodes to obtain example-level representations $\bar{\mathbf{H}}_{\mathbf{k}} \in \mathbb{R}^{(k+1) \times d_{\text{node}}}$. These are projected through an MLP to yield embeddings $\mathbf{Z}$, from which we compute head-specific pairwise similarities:

$$\mathbf{S} = \frac{(\mathbf{W}_q \mathbf{Z})(\mathbf{W}_k \mathbf{Z})^\top}{\sqrt{d_{\text{node}}/H}} \in \mathbb{R}^{H \times (k+1) \times (k+1)} \quad (5)$$

The attention bias for position pair $(i, j)$ is set as $\mathbf{A}[:, i, j] = \mathbf{S}[:, \lfloor i/2 \rfloor, \lfloor j/2 \rfloor]$, so key–value pairs within the same example share the same bias. Since this bias is derived from content rather than positional indices, it naturally generalises to arbitrary example counts at inference.

For key–value distinction, learnable offset vectors $\pm\mathbf{r}$ are added to key $(+\mathbf{r})$ and value $(-\mathbf{r})$ tokens. Pseudo-code is provided in Algorithm 1 (Appendix B).

## 3. Experiments

**Setup.** We evaluate on air quality data from two Chinese regions: BTHSA (Beijing-Tianjin-Hebei, 228 stations) and YRD (Yangtze River Delta, 127 stations), with 13 features per station (Wang et al., 2025b). Each operator corresponds to a prediction horizon $\Delta t$. Classical single-operator baselines train on fixed $\Delta t \in \{1, 4, 12, 24\}$h without examples ($k = 0$). Multi-operator ICON trains with $\Delta t \sim \text{Uniform}[1, 24]$h and up to $k \in \{1, 2, 5\}$ examples. All models use 90,000 training steps. We evaluate with up to 100 examples and report RMSE over all stations. Full setup details, including data processing, graph construction, model configuration, and optimisation, are provided in Appendix C.

### 3.1. Example Cardinality Generalisation

We evaluate how cardinality generalisation varies with operator complexity across $\Delta t \in \{1, 4, 12, 24\}$h. Models trained with maximum $k \in \{1, 2, 5\}$ examples maintain stable performance when evaluated with up to 100 examples, confirming that example-aware positional encoding generalises beyond training counts. For simple operators ($\Delta t = 1, 4$h), classical single-operator learning achieves lower RMSE (Appendix E), as dedicated models can specialise effectively. For complex operators ($\Delta t = 24$h, Figure 2 left), ICON with operator diversity surpasses the baseline with sufficient examples, with the advantage becoming more pronounced at larger $\Delta t$. The contrast is especially clear for $O_3$, where ICON with operator diversity exhibits a substantial RMSE drop with just one example, while the single-operator baseline remains flat.

### 3.2. Out-of-Distribution Operator Extrapolation

Having established cardinality generalisation for in-distribution operators, we evaluate extrapolation to an unseen operator: models trained with $\Delta t \in [1, 24]$ are tested at $\Delta t = 48$ (out-of-distribution). We vary maximum example count $k \in \{1, 2, 5\}$ and evaluate performance as a function of validation example count (Figure 2 centre). For comparison, we include the classical single-operator baseline ($\Delta t = 24$, trained without examples). It shows flat performance regardless of example count during inference.

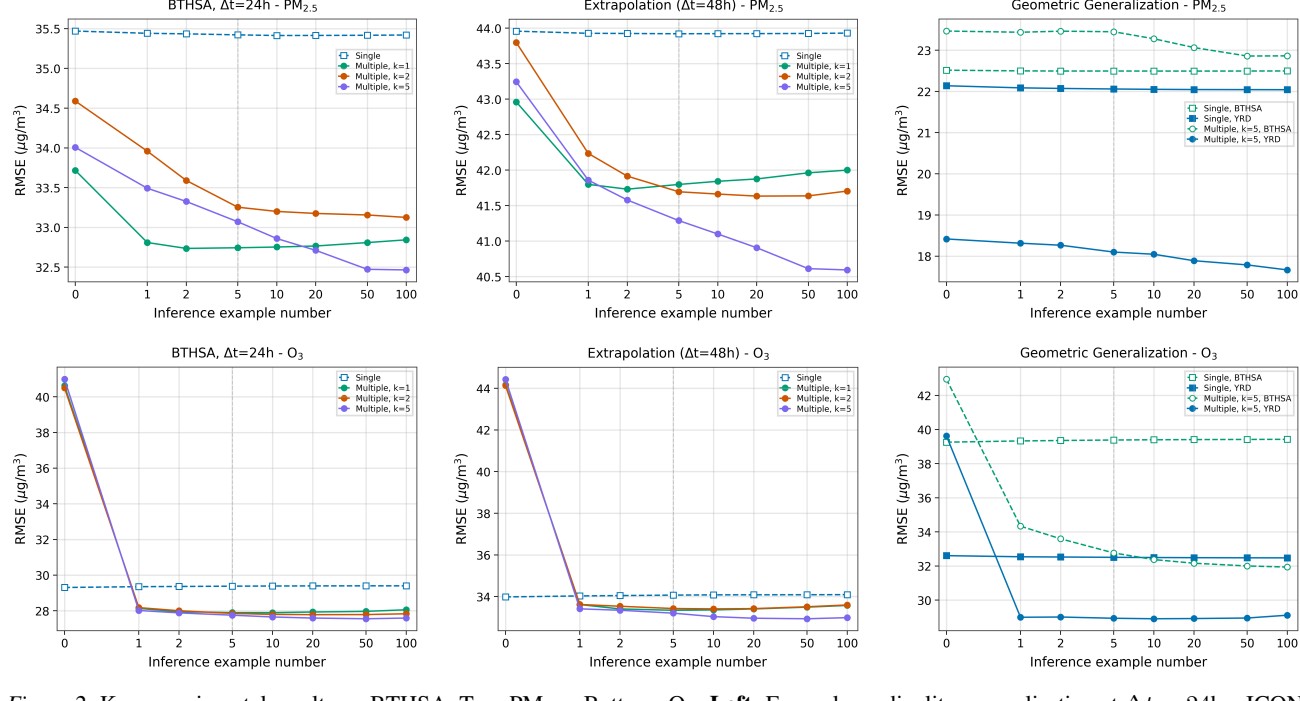

*Figure 2.* Key experimental results on BTHSA. Top: PM$_{2.5}$. Bottom: O$_3$. **Left**: Example cardinality generalisation at $\Delta t = 24$h—ICON with operator diversity surpasses the single-operator baseline with sufficient examples (full results across all $\Delta t$ in Appendix E). **Centre**: OOD operator extrapolation to $\Delta t = 48$h—single-operator shows flat curves while ICON models improve with examples. **Right**: Geometric generalisation—models trained on BTHSA transfer to YRD ($\Delta t = 24$h) without fine-tuning, with moderate gap for PM$_{2.5}$ and minimal gap for O$_3$.

Under the current paradigm, ICON with operator diversity shows markedly different behaviour: for O$_3$, all example-trained models exhibit a substantial RMSE drop with just one example; for PM$_{2.5}$, error continues to decrease with additional examples. The model trained with $k = 5$ achieves the best extrapolation, with RMSE generally decreasing up to 100 examples. Since multi-operator training presents a different operator each batch, the model must rely on examples to identify the operator in play, creating a natural incentive to extract operator-relevant information rather than memorising the training distribution.

### 3.3. Geometric Generalisation

To assess transferability across spatial domains, we conduct cross-region experiments. We train models separately on BTHSA (228 stations) and YRD (127 stations), then evaluate both on YRD at $\Delta t = 24$h without fine-tuning. This tests whether a model trained on one region can generalise to a different graph topology. Figure 2 (right) shows the results.

For PM$_{2.5}$, BTHSA-trained ICON models transferred to YRD show a moderate performance gap compared to YRD-native models, but maintain stable performance across example counts, suggesting that learned representations transfer across graph topologies. For O$_3$, the transfer gap is notably

smaller; the BTHSA-trained ICON model with sufficient examples even surpasses the classical single-operator baseline trained natively on YRD, suggesting that multi-operator in-context learning may compensate for domain mismatch.

Ablation studies examining whether single-operator in-context learning can also benefit from examples are provided in Appendix G.

## 4. Conclusion

We presented GICON (Graph In-Context Operator Network), combining graph message passing with example-aware positional encoding to enable in-context operator learning on irregularly sampled spatiotemporal systems. Through controlled experiments on air quality prediction, we demonstrated that in-context operator learning with operator diversity outperforms classical single-operator learning on complex tasks, with examples becoming increasingly informative as count grows—extending even to out-of-distribution operators. GICON further achieves geometric generalisation across regions with different graph structures and spatial geometries, and cardinality generalisation from 0–5 training examples to 100 at inference with generally improving performance. These results suggest GICON as a promising approach for generalizable spatiotemporal prediction.

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

# A. Detailed Method Description

## A.1. Problem Setup

We consider a forward problem defined on domain $\Omega \subseteq \mathbb{R}^d$ where $d$ is the spatial dimension ($d = 2$ in this work), with state represented by $\mathbf{u}(\mathbf{x}, t) : \Omega \times [0, T] \to \mathbb{R}^c$, where $c$ is the number of channels (e.g., temperature, pressure, velocity components). Our goal is to predict the future state $\mathbf{u}_{t+\Delta t}$ given historical observations.

**Graph Representation.** To achieve better spatial symmetry and handle irregular spatial domains naturally, we represent the spatial domain $\Omega$ as a graph $\mathcal{G} = (\mathcal{V}, \mathcal{E})$, where $\mathcal{V}$ is the set of nodes and $\mathcal{E}$ is the set of edges. Each node $v_i \in \mathcal{V}$ corresponds to a spatial location $\mathbf{x}_i \in \Omega$, and edges $e_{ij} \in \mathcal{E}$ connect nodes based on spatial proximity or other physical connectivity criteria. This graph structure naturally accommodates irregular meshes, non-uniform spatial sampling, and complex boundary geometries that are common in real-world weather data. We denote the node features at time $t$ as $\mathbf{u}_t = \mathbf{u}(\cdot, t) \in \mathbb{R}^{|\mathcal{V}| \times c}$, where $|\mathcal{V}|$ is the number of nodes in the graph. Each row $\mathbf{u}_t^{(i)} \in \mathbb{R}^c$ represents the $c$-dimensional feature vector at node $v_i$.

However, compared to VICON (Cao et al., 2026) which operates on dense pixel grids, our graph-based representation introduces a key challenge: the sparse discrete node sampling breaks the Markovian property. In dense continuous representations of first-order-in-time systems, a single snapshot at time $t$ captures sufficient local spatial information to infer the next state. In contrast, sparse graph nodes only observe the state at discrete locations, losing the fine-grained spatial structure between nodes. Consequently, a single snapshot $\mathbf{u}_t$ becomes insufficient to uniquely determine the future evolution. To recover the temporal dynamics, we must rely on a sequence of historical frames $\{\mathbf{u}_{t-\tau+1}, \ldots, \mathbf{u}_{t-1}, \mathbf{u}_t\}$ (where $\tau$ is the historical window size) from which the temporal evolution patterns can be inferred.

**Operator Learning Framework.** We assume that the dynamics of the system can be modelled by a family of evolution operators $\{\mathcal{F}_{\Delta t}\}_{\Delta t > 0}$, such that

$$\mathbf{u}_{t+\Delta t} = \mathcal{F}_{\Delta t}(\mathbf{u}_{t-\tau+1}, \ldots, \mathbf{u}_{t-1}, \mathbf{u}_t). \tag{6}$$

The operator $\mathcal{F}_{\Delta t}$ is inferred from a set of contextual examples that share the same time gap $\Delta t$. Unlike traditional numerical solvers that discretise governing equations, our approach learns the solution operator directly from observation data, enabling efficient predictions without requiring explicit knowledge of the underlying physics.

**In-Context Learning Setup.** Following the in-context learning paradigm established in ICON (Yang et al., 2023;

2025; Yang & Osher, 2024), we aim to develop a model that can adapt to new scenarios by leveraging a set of contextual examples. Each example consists of a pair $(\mathbf{k}, \mathbf{v})$, where $\mathbf{k}$ represents the key and $\mathbf{v}$ represents the value. In general, keys and values are input-output function pairs; in our spatiotemporal setting, keys correspond to historical observations and values to future states. In our graph-based setting, we denote $\mathbf{k}^{(j)} = \mathbf{u}_{t_j - \tau + 1 : t_j}^{(j)}$ and $\mathbf{v}^{(j)} = \mathbf{u}_{t_j + \Delta t}^{(j)}$, forming the example set $\mathcal{D} = \{(\mathbf{k}^{(j)}, \mathbf{v}^{(j)})\}_{j=1}^{N_d}$, where $N_d$ is the number of examples. These examples provide contextual information about the specific dynamics governing the current prediction task. The model processes both the examples and the query key $\mathbf{k}_{\text{query}} = \mathbf{u}_{t-\tau+1:t}$ to produce the prediction $\tilde{\mathbf{v}}_{\text{query}} = \hat{\mathbf{u}}_{t+\Delta t}$, allowing it to adapt its predictions based on the provided context without requiring gradient-based fine-tuning. This formulation performs "next function prediction" analogous to "next token prediction" in language models.

**Training and Operator Consistency.** For efficient training parallelisation, we employ a causal attention mask that exploits varying numbers of examples to compute all predictions in a single forward pass. The training loss is computed as the mean squared error (MSE) between predicted and ground truth values at both key positions and the query position. An important constraint is that all example pairs in the same sequence must be formed with the same operator mapping (i.e., the same $\Delta t$), ensuring the model learns a coherent operator from the context. However, operators can vary across different training sequences, enabling generalisation across multiple operator types.

## A.2. Retrieval of Examples

To effectively utilise the contextual examples in a scalable manner, we implement a retrieval mechanism that selects the most relevant examples from a potentially large pool of examples. This retrieval process serves two purposes: (1) it reduces computational costs by avoiding processing all available examples, and (2) it is expected to improve prediction quality by focusing on contextually relevant patterns.

**Feature Extraction.** Given a query key $\mathbf{k}_{\text{query}} = \mathbf{u}_{t-\tau+1:t}$ and an example pool $\mathcal{D} = \{(\mathbf{k}^{(j)}, \mathbf{v}^{(j)})\}_{j=1}^{N_d}$, we first extract representative features from the key sequences. Rather than using all $\tau$ frames in the key, we focus on the last $\tau_r$ frames (where $\tau_r \leq \tau$) to capture the most recent dynamics. For each key sequence, we compute a pooled feature representation by aggregating spatial and temporal information. Specifically, for a key $\mathbf{k}$ consisting of frames $\{\mathbf{u}_{t-\tau_r+1}, \ldots, \mathbf{u}_t\}$, we apply global average pooling over the node dimension for each frame, resulting in temporal feature vectors, which are then concatenated or further aggregated to form a fixed-size representation $\mathbf{z} \in \mathbb{R}^d$.

**Similarity-Based Retrieval.** We compute the similarity between the query feature $\mathbf{z}_{\text{query}}$ and each example feature $\mathbf{z}_{\text{ex}}^{(j)}$ using cosine similarity:

$$s_j = \frac{\mathbf{z}_{\text{query}}^{\top}\mathbf{z}_{\text{ex}}^{(j)}}{\|\mathbf{z}_{\text{query}}\|_2\|\mathbf{z}_{\text{ex}}^{(j)}\|_2}. \tag{7}$$

For efficient nearest neighbour search over large example pools, we employ the FAISS library (Douze et al., 2025), which provides optimised implementations for approximate nearest neighbour search in high-dimensional spaces. FAISS enables us to scale to millions of examples while maintaining sub-linear query time through techniques such as product quantisation and inverted file indexing.

**Two-Stage Selection.** Based on the computed similarity scores $\{s_j\}_{j=1}^{N_d}$, we first select the top-$K$ examples with the highest scores:

$$\mathcal{D}_K = \{(\mathbf{k}^{(j)}, \mathbf{v}^{(j)}) : j \in \text{top-}K(\{s_j\})\}, \tag{8}$$

where top-$K(\cdot)$ returns the indices of the $K$ largest values. During training, to introduce randomness and prevent overfitting to specific example combinations, we randomly sample $k$ examples from $\mathcal{D}_K$ (where $k \leq K$) in each epoch to form the actual training context. This two-stage approach combines the benefits of similarity-based retrieval with stochastic data augmentation.

### A.3. Network Architecture

Our Graph In-Context Operator Network (GICON) integrates graph neural network message passing with in-context operator learning through an architecture that processes sequences of graph-structured data. Given the sampled example set $\mathcal{D}_k = \{(\mathbf{k}^{(j)}, \mathbf{v}^{(j)})\}_{j=1}^{k}$ and the query $\mathbf{k}_{\text{query}}$, GICON processes these as an interleaved sequence to predict $\tilde{\mathbf{v}}_{\text{query}}$.

**Input Projection and Sequence Construction.** Each key $\mathbf{k}$ and the query $\mathbf{k}_{\text{query}}$ consist of $\tau$ historical frames, which we flatten along the feature dimension and project to a common node embedding space via $\text{proj}_{\mathbf{k}}$ : $\mathbb{R}^{\tau \times d_{\text{in}}} \to \mathbb{R}^{d_{\text{node}}}$. Each value $\mathbf{v}$ (a single frame) is projected via $\text{proj}_{\mathbf{v}}$ : $\mathbb{R}^{d_{\text{in}}} \to \mathbb{R}^{d_{\text{node}}}$. We then construct an interleaved sequence by alternating keys and values: $[\mathbf{k}^{(1)}, \mathbf{v}^{(1)}, \mathbf{k}^{(2)}, \mathbf{v}^{(2)}, \ldots, \mathbf{k}^{(k)}, \mathbf{v}^{(k)}, \mathbf{k}_{\text{query}}]$, resulting in a sequence of length $2k + 1$.

**Graph Message Passing with In-Context Learning.** The core of GICON consists of $L$ stacked layers, each combining spatial message passing with in-context learning. For each layer $\ell$, we maintain hidden states $\mathbf{h}^{(\ell)} \in \mathbb{R}^{(2k+1) \times |\mathcal{V}| \times d_{\text{node}}}$.

Each GICON layer performs two sequential operations:

1. **Spatial Message Passing:** For each sequence position $t$ (in parallel across all positions), we aggregate information from neighbouring nodes in the graph. Given hidden states $\mathbf{h}_t^{(\ell)} \in \mathbb{R}^{|\mathcal{V}| \times d_{\text{node}}}$ at sequence position $t$, we compute messages for each node $i$:

$$\mathbf{m}_{t,i} = \sum_{j \in \mathcal{N}(i)} \text{MLP}_{\text{msg}}([\mathbf{h}_{t,i}^{(\ell)}, \mathbf{h}_{t,j}^{(\ell)}, \mathbf{e}_{ij}]) \tag{9}$$

where $\mathcal{N}(i)$ denotes the neighbours of node $i$, and $\mathbf{e}_{ij}$ is the edge feature. The updated hidden states incorporate these aggregated messages: $\tilde{\mathbf{h}}_{t,i}^{(\ell)} = \mathbf{h}_{t,i}^{(\ell)} + \mathbf{m}_{t,i}$.

2. **Per-Node In-Context Learning:** For each node $i$ (in parallel across all nodes), we apply a transformer across the sequence dimension to enable in-context learning. We rearrange the hidden states to node-centric view $\tilde{\mathbf{h}}_i^{(\ell)} \in \mathbb{R}^{(2k+1) \times d_{\text{node}}}$ and apply:

$$\mathbf{h}_i^{(\ell+1)} = \text{Transformer}(\tilde{\mathbf{h}}_i^{(\ell)}, \text{mask}_{\text{causal}}) \tag{10}$$

where $\text{mask}_{\text{causal}}$ ensures autoregressive prediction by preventing positions from attending to future positions in the sequence.

This design enables each node to learn from spatial neighbours while simultaneously performing in-context learning across contextual examples, combining the geometric flexibility of GNNs with the adaptability of in-context learning.

**Positional Encoding for Example Distinction.** A critical challenge in GICON is distinguishing between (1) different contextual examples versus the query, and (2) keys versus values within the sequence. We introduce two complementary positional encoding strategies:

For inter-example distinction, we introduce an example-aware attention bias $\mathbf{A} \in \mathbb{R}^{H \times (2k+1) \times (2k+1)}$ that operates at the example level. We first extract key tokens (at even sequence positions) and pool them across nodes to obtain example-level representations $\bar{\mathbf{H}}_{\mathbf{k}} \in \mathbb{R}^{(k+1) \times d_{\text{node}}}$. These are projected through an MLP to yield example embeddings $\mathbf{Z} = \text{MLP}(\bar{\mathbf{H}}_{\mathbf{k}}) \in \mathbb{R}^{(k+1) \times d_{\text{node}}}$, from which we compute head-specific pairwise similarities:

$$\mathbf{S} = \frac{(\mathbf{W}_q \mathbf{Z})(\mathbf{W}_k \mathbf{Z})^{\top}}{\sqrt{d_{\text{node}}/H}} \in \mathbb{R}^{H \times (k+1) \times (k+1)} \tag{11}$$

The attention bias for each position pair $(i, j)$ is then set as $\mathbf{A}[:, i, j] = \mathbf{S}[:, \lfloor i/2 \rfloor, \lfloor j/2 \rfloor]$, so that each value token inherits the bias of its corresponding key within the same example. This content-aware bias encourages attending to examples with similar keys, and since it is derived from

content rather than positional indices, it naturally generalises to any number of examples at inference time.

For key-value distinction, we add learnable offset vectors $\pm\mathbf{r}$ to the token embeddings, where key tokens receive $+\mathbf{r}$ and value tokens receive $-\mathbf{r}$, creating a separation between keys and values.

This multi-level positional encoding scheme enables GICON to effectively leverage the hierarchical structure of in-context operator learning on graphs, distinguishing both the example-query relationship and the key-value pairing within each example. Detailed pseudo-code is provided in Algorithm 1 (Appendix B).

## B. Positional Encoding Algorithms

---

**Algorithm 1** Positional Encoding for GICON

---

**Require:** Hidden states $\mathbf{H} \in \mathbb{R}^{|\mathcal{V}| \times (2k+1) \times d}$, number of attention heads $H$

1: **— Inter-Example Distinction: Example-Aware Attention Bias —**
2: {Returns bias $\mathbf{A} \in \mathbb{R}^{H \times (2k+1) \times (2k+1)}$}
3: Extract key tokens at even positions: $\mathbf{H_k} \leftarrow \mathbf{H}[:, 0 :: 2, :] \in \mathbb{R}^{|\mathcal{V}| \times (k+1) \times d}$
4: Pool across nodes: $\bar{\mathbf{H}}_\mathbf{k} \leftarrow \mathrm{mean}(\mathbf{H_k}, \dim = \mathrm{nodes}) \in \mathbb{R}^{(k+1) \times d}$
5: Project to example embeddings: $\mathbf{Z} \leftarrow \mathrm{MLP}(\bar{\mathbf{H}}_\mathbf{k}) \in \mathbb{R}^{(k+1) \times d}$
6: Compute head-specific query/key: $\mathbf{Q}_z \leftarrow \mathbf{W}_q\mathbf{Z}$, $\mathbf{K}_z \leftarrow \mathbf{W}_k\mathbf{Z}$ {$\mathbf{W}_q, \mathbf{W}_k \in \mathbb{R}^{d \times d}$}
7: Compute pairwise similarities: $\mathbf{S} \leftarrow \mathbf{Q}_z\mathbf{K}_z^\top / \sqrt{d/H} \in \mathbb{R}^{H \times (k+1) \times (k+1)}$
8: Initialize bias: $\mathbf{A} \leftarrow \mathbf{0}^{H \times (2k+1) \times (2k+1)}$
9: **for** each position pair $(i, j)$ where $\lfloor i/2 \rfloor \le k$ and $\lfloor j/2 \rfloor \le k$ **do**
10: $\quad \mathbf{A}[:, i, j] \leftarrow \mathbf{S}[:, \lfloor i/2 \rfloor, \lfloor j/2 \rfloor]$ {Key-value pairs share same bias}
11: **end for**

12: **— Key-Value Distinction: Input Mode —**
13: {Learnable KV vector $\mathbf{r} \in \mathbb{R}^d$}
14: Create KV masks: $\mathbf{M_k} \leftarrow (t \bmod 2 = 0)$, $\mathbf{M_v} \leftarrow (t \bmod 2 = 1)$ for $t \in [0, 2k]$
15: Apply symmetric offsets: $\tilde{\mathbf{H}} \leftarrow \mathbf{H} + \mathbf{M_k} \cdot \mathbf{r} - \mathbf{M_v} \cdot \mathbf{r}$
16: **return** $\tilde{\mathbf{H}}$ {Keys shifted by $+\mathbf{r}$, values by $-\mathbf{r}$}

---

## C. Training Setup

### C.1. Datasets

We evaluate on air quality monitoring datasets from two major Chinese regions, following the benchmark established by (Wang et al., 2025b). The BTHSA dataset covers Beijing-Tianjin-Hebei and Surrounding Areas ($\sim$430,000 km$^2$, 28 cities) with 228 monitoring stations, while the YRD dataset covers the Yangtze River Delta ($\sim$270,000 km$^2$) with 127 stations. Together, these provide 70,128 hours of observations spanning 2016–2023.

Each station records 10 raw features: 2 air quality variables (hourly PM$_{2.5}$ and O$_3$ concentrations from CNEMC) and 8 meteorological variables (wind components, temperature, precipitation, and surface pressure from ERA5 reanalysis). We derive 3 additional features: hour of day as a temporal encoding, and wind speed/direction computed from wind components ($u_{100}$, $v_{100}$) using MetPy (May et al., 2022), yielding 13 input features per station. Spatial connections between stations are established using a 200km geodesic distance threshold; edges are further filtered by removing connections where intervening terrain exceeds 1200m above source or destination altitude, modelling physical barriers to pollutant transport. Each edge carries attributes of geodesic distance and direction. We use 2016–2022 as the example retrieval pool, 2017–2022 for training, and 2023 for testing. Both datasets are publicly available at https://zenodo.org/records/15614907.

### C.2. Training and Evaluation Protocol

Each operator corresponds to temporal evolution with time gap $\Delta t$, and we set the historical window size $\tau = \tau_r = 24$ (i.e., each key consists of 24 hourly frames). The classical single-operator learning baseline trains on a fixed $\Delta t \in \{1, 4, 12, 24\}$ hours without contextual examples ($k = 0$). For the ICON paradigm, we primarily use multi-operator training with $\Delta t$ sampled uniformly from $[1, 24]$ hours, with maximum example counts $k \in \{1, 2, 5\}$; we additionally examine single-operator in-context learning ($k \in \{1, 2, 5\}$) as an ablation study (Appendix G). All models are trained for the same number of steps (90,000), but since each valid $\Delta t$ value generates a separate set of training sequences, single-operator models see each sample $\sim$24$\times$ more often during training, while multi-operator models trade per-sample repetition for operator diversity. Models are evaluated with up to 100 examples to assess cardinality generalisation. Note that all models, including single-operator models and those trained without contextual examples ($k = 0$), are provided with examples during evaluation; this allows us to test whether models trained on a single operator or without examples can still leverage in-context examples at inference. We adopt Root Mean Square Error (RMSE) as our primary evaluation metric, computed over all stations as $\mathrm{RMSE} = \sqrt{\frac{1}{N} \sum_{i=1}^N (\hat{y}_i - y_i)^2}$.

### C.3. Model Configuration

*Table 1.* GICON model configuration.

| Component | Configuration |
|---|---|
| *Graph In-Context Operator Network* | |
| GICON layers | 3 |
| *Message Passing Neural Network* | |
| Node dimension | 128 |
| Edge dimension | 128 |
| Message dimension | 256 |
| *In-Context Transformer* | |
| Attention heads | 4 |
| Feed-forward dimension | 512 |
| Dropout | 0.1 |
| Normalization | RMSNorm* |

*(Zhang & Sennrich, 2019)

*Table 2.* Optimizer and learning rate schedule.

| Parameter | Value |
|---|---|
| *Muon Optimizer (Jordan et al., 2024)* | |
| Learning rate | $1 \times 10^{-4}$ |
| Weight decay | 0.01 |
| Gradient clipping | 1.0 |
| *Cosine Decay Schedule* | |
| Warmup | 10% of training |
| Decay period | 100% of training |
| End learning rate factor | 0.1 |

Table 1 summarises the GICON architecture, and Table 2 details the optimisation setup.

**Loss Function.** The training objective is the mean squared error (MSE) computed over predicted pollution concentrations at all key positions and the query position in the sequence. Given $k$ in-context examples and one query, the causal attention mask enables the model to produce predictions $\tilde{\mathbf{v}}^{(1)}, \tilde{\mathbf{v}}^{(2)}, \ldots, \tilde{\mathbf{v}}^{(k)}$ at example key positions and $\tilde{\mathbf{v}}_{\text{query}}$ at the query position in a single forward pass. We extract the pollution channels (the last two channels) from each predicted value as $\tilde{\mathbf{p}}^{(j)}$ and similarly $\mathbf{p}^{(j)}$ from the ground truth $\mathbf{v}^{(j)}$. The loss is

$$\mathcal{L} = \text{MSE}\left(\tilde{\mathbf{p}}, \mathbf{p}\right), \tag{12}$$

where $\tilde{\mathbf{p}} = [\tilde{\mathbf{p}}^{(1)}; \tilde{\mathbf{p}}^{(2)}; \ldots; \tilde{\mathbf{p}}^{(k)}; \tilde{\mathbf{p}}_{\text{query}}]$ and $\mathbf{p} = [\mathbf{p}^{(1)}; \mathbf{p}^{(2)}; \ldots; \mathbf{p}^{(k)}; \mathbf{p}_{\text{label}}]$ denote the concatenation of predicted and ground-truth pollution values across all $k$ in-context examples and the query.

## D. Training Dynamics

This section provides training dynamics for both datasets, evaluated at $\Delta t = 24$h. Single-operator models exhibit clear overfitting: validation performance deteriorates despite continued improvement on training loss. In contrast, ICON with operator diversity exhibits negligible overfitting across the entire training process (Figure 3). All comparison experiments use checkpoints at 90,000 training steps to ensure a fair comparison across all models under the same training steps and dataset.

Figure 4 shows training dynamics for the single-operator ablation with varying maximum example count $k$. All settings exhibit overfitting, with $k = 0$ achieving the best validation performance across all single-operator settings.

## E. Example Cardinality Generalisation

This section provides full example cardinality generalisation results, complementing the $\Delta t = 24$h results in the main text (Figure 2).

### E.1. BTHSA: Simple to Moderate Operators

For simple operators ($\Delta t = 1, 4$h) and the moderate operator ($\Delta t = 12$h), classical single-operator learning achieves lower RMSE, as dedicated models can specialise effectively. At $\Delta t = 12$h, ICON with operator diversity begins to approach and surpass the baseline with sufficient examples (Figure 5).

### E.2. YRD Dataset

The same trend is observed on the YRD dataset: for simple operators ($\Delta t = 1, 4$h), classical single-operator learning achieves lower RMSE (Figure 6), while for complex operators ($\Delta t = 12, 24$h), ICON with operator diversity surpasses the baseline with sufficient examples (Figure 7).

## F. Related Work

### F.1. Operator Learning

Operator learning (Chen & Chen, 1995a;b; Lu et al., 2021; Li et al., 2021) addresses the challenge of approximating operators $\mathcal{G} : \mathcal{U} \to \mathcal{V}$, where $\mathcal{U}, \mathcal{V}$ are function spaces. These operators typically model solutions of physical systems and differential equations, representing the map from initial/boundary conditions and parameters to equation solutions. Early solution approximation approaches—including deep learning for high-dimensional PDEs (E et al., 2017; Han et al., 2018), the Deep Ritz Method (E & Yu, 2018), and PDE-Net (Long et al., 2018)—train neural networks to approximate the solution of a specific PDE instance but require retraining for each new instance.

DeepONets (Lu et al., 2021; 2022) use branch and trunk networks to separately process inputs and query points, and have been extended to handle multiple input parametric functions in MIONet (Jin et al., 2022). Fourier Neural Operators (FNOs) (Li et al., 2021; Kovachki et al., 2023) leverage kernel integration approximations for PDE solutions by processing features in Fourier space. These methods have been further extended to incorporate equation information (Wang et al., 2021), multiscale features (Zhang et al., 2024a; Wen et al., 2022), different input/output meshes (Zhang et al., 2023; Hao et al., 2023), complex geometries (Li et al., 2023), and distributed learning (Zhang et al., 2024b). Applications include weather prediction (Kurth et al., 2023), geosciences (Jiang et al., 2024), biology and medicine (Yin et al., 2024), and uncertainty quantification (Moya et al., 2025).

However, these approaches are typically designed to learn a single operator, requiring retraining when encountering different types of PDEs or time-step sizes. Pretraining-based approaches such as Poseidon (Herde et al., 2024) and LeMON (Sun et al., 2024) address this through multi-operator pretraining followed by fine-tuning. Beyond fine-tuning, in-context approaches aim to generalise across operators without task-specific optimisation.

### F.2. Graph Neural Networks for Operator Learning

Graph Neural Networks (GNNs) have emerged as an effective framework for learning physics simulations by leveraging relational inductive biases (Battaglia et al., 2018). Unlike grid-based methods, GNNs naturally handle irregular geometries and sparse interactions through message passing on graph structures.

GNNs have demonstrated strong performance in learning complex physical dynamics (Sanchez-Gonzalez et al., 2020), including particle-based systems and fluid simulations. This approach has been extended to mesh-based simulations through MeshGraphNets (Pfaff et al., 2021) and to PDE solving through message-passing neural networks (Brandstetter et al., 2022). GNN-based neural operators include the Graph Neural Operator (GNO) (Li et al., 2020b), Multipole GNO (Li et al., 2020a), Geometry-Informed Neural Operator (GINO) (Li et al., 2023), and Region Interaction GNO (RIGNO) (Mousavi et al., 2025). While these methods excel at handling irregular geometries, they typically learn a fixed operator for a specific PDE system.

### F.3. In-Context Operator Networks

In-Context Operator Networks (ICONs) (Yang et al., 2023), inspired by in-context learning in large language models (Radford et al., 2019; Brown et al., 2020), learn an *operator learner* that infers new operators from prompted input-output pairs without weight updates. Several extensions have advanced the core framework: ICON-LM (Yang et al., 2025) incorporates autoregressive next-function prediction; (Yang & Osher, 2024) demonstrated generalisation to unseen PDE forms; and GenICON (Zhang et al., 2025) introduces probabilistic modelling and uncertainty quantification.

The paradigm has also been applied beyond differential equations, including optimal execution in finance (Meng et al., 2025) and optimal transport (Cole et al., 2026). Theoretically, it is supported by empirical robustness analysis (Liu et al., 2023), generalization bounds proving that task diversity is necessary and sufficient for out-of-domain generalization (Cole et al., 2024), and connections to gradient descent in function spaces (Mishra et al., 2025). Parallel approaches include PROSE (Liu et al., 2024b;a), which conditions on symbolic equation representations, (Chen et al., 2024), which combines unsupervised pretraining with in-context learning, and Zebra (Serrano et al., 2025) and ENMA (Koupaï et al., 2025), which employ autoregressive latent-space generation for parametric PDEs.

Vanilla ICONs face quadratic complexity from point-wise function representation, limiting them to 1D problems or sparse 2D sampling. VICON (Cao et al., 2026) addresses this through patch-based sequences, enabling efficient dense 2D processing. However, existing methods share two practical limitations: grid-based representations that restrict applicability to irregular geometries, and fixed example counts during training that prevent cardinality generalisation at inference.

## G. Ablation: Single-Operator Learning with Examples

The main-text experiments show that ICON with operator diversity substantially outperforms classical single-operator learning on complex tasks. A natural question arises: can single-operator models also benefit from examples within the in-context operator learning paradigm? We investigate this through ablation studies that examine the effect of examples when operator diversity is absent. Training dynamics show that single-operator settings exhibit more overfitting than multi-operator training (Appendix D).

To complement the cardinality results in the main text (Section 3.1), we examine how models trained with single-operator examples scale with example count (Figure 8). Providing examples during inference does improve performance over the zero-example baseline for all models trained with single-operator examples. In particular, the model trained with $k = 5$ exhibits continued improvement as example count increases and approaches the classical single-operator baseline, slightly surpassing it for $PM_{2.5}$ at 100 examples. To further probe whether examples pro-

vide meaningful information, we compare inference with high-quality examples versus random Gaussian noise. For the model trained without contextual examples ($k = 0$), noise examples perform comparably to high-quality ones, suggesting the model has learned to ignore example content entirely. In contrast, models trained with examples show performance degradation when given noise, indicating sensitivity to example content.

Together, the cardinality scaling and noise sensitivity results show some signal that single-operator models can learn to utilise examples, particularly when trained with sufficient example count ($k = 5$). However, the improvement is limited—the margin over the classical baseline is only observed for $PM_{2.5}$ and not for $O_3$—and compared to ICON with operator diversity, single-operator models require more examples to compensate for generally poorer performance and are more prone to overfitting (Appendix D). How exactly examples are utilised within the model's forward pass, and how to further exploit in-context examples in single-operator scenarios, remain open questions.

# Supplementary Figures

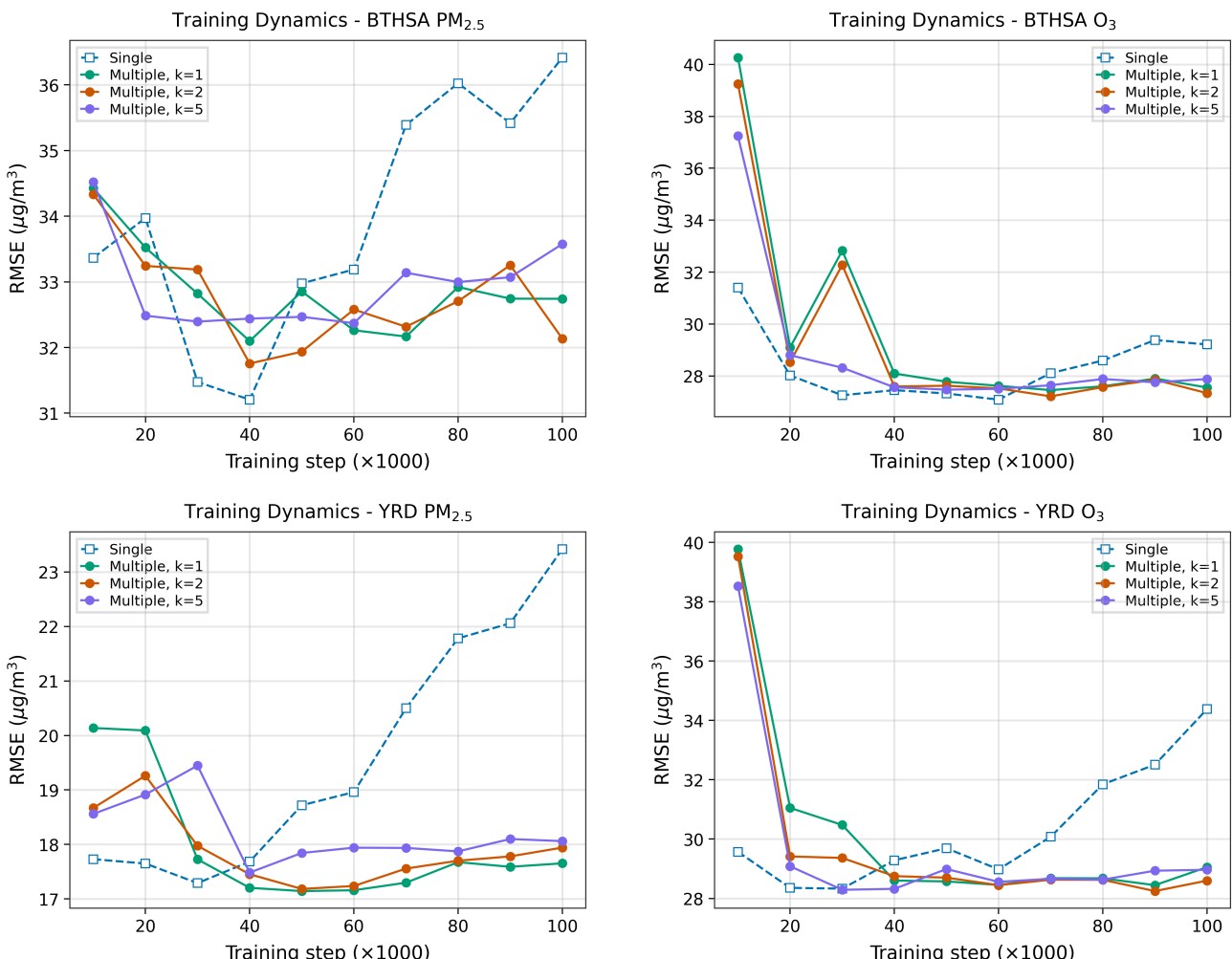

*Figure 3.* Training dynamics: multi-operator vs. single-operator comparison at $\Delta t = 24$h. Left: PM$_{2.5}$. Right: O$_3$. Top: BTHSA. Bottom: YRD. ICON with operator diversity exhibits negligible overfitting while single-operator models overfit.

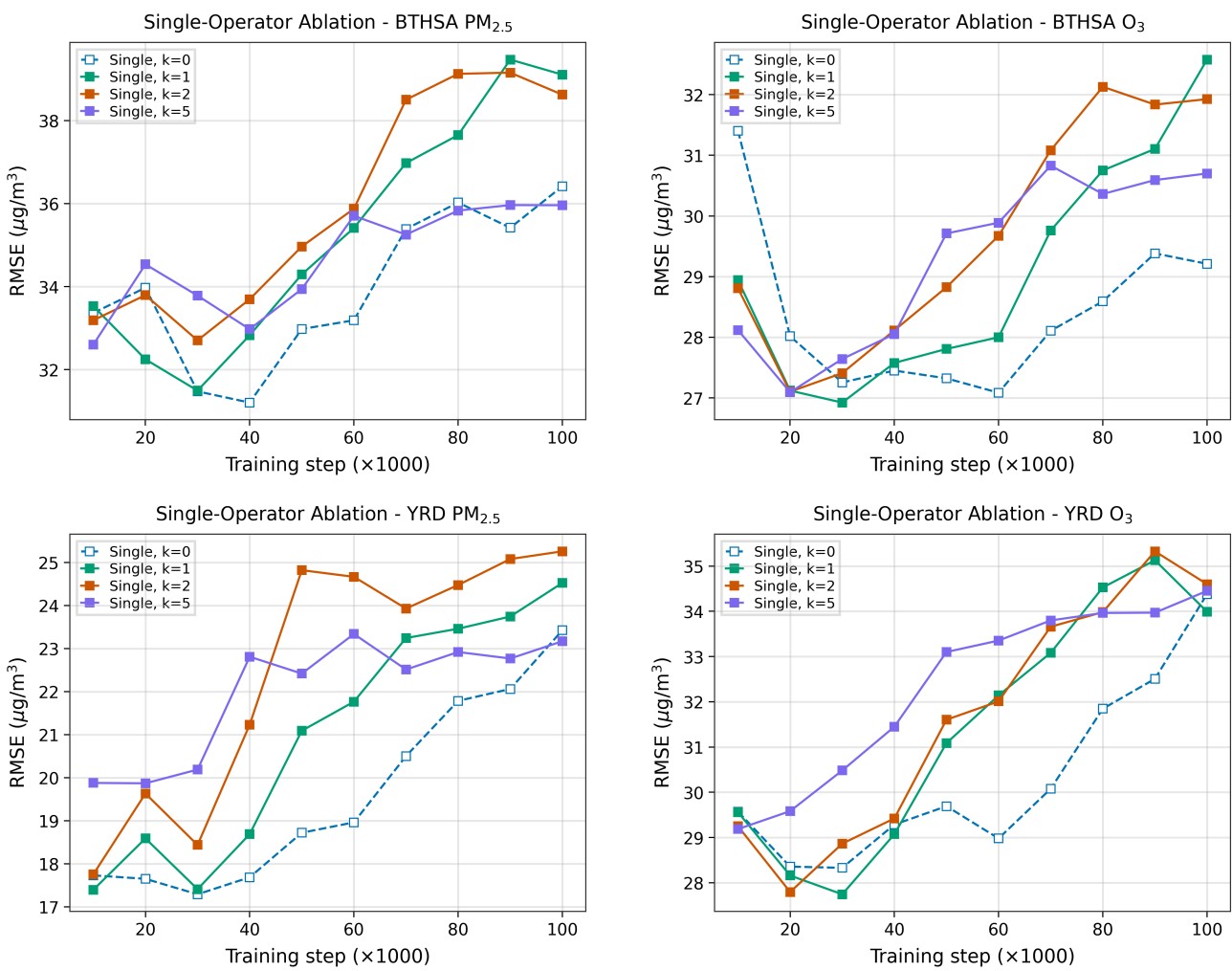

*Figure 4.* Training dynamics: single-operator ablation at $\Delta t = 24$h with varying $k$. Left: PM$_{2.5}$. Right: O$_3$. Top: BTHSA. Bottom: YRD. All settings exhibit overfitting, with $k = 0$ achieving the best validation performance across all single-operator settings.

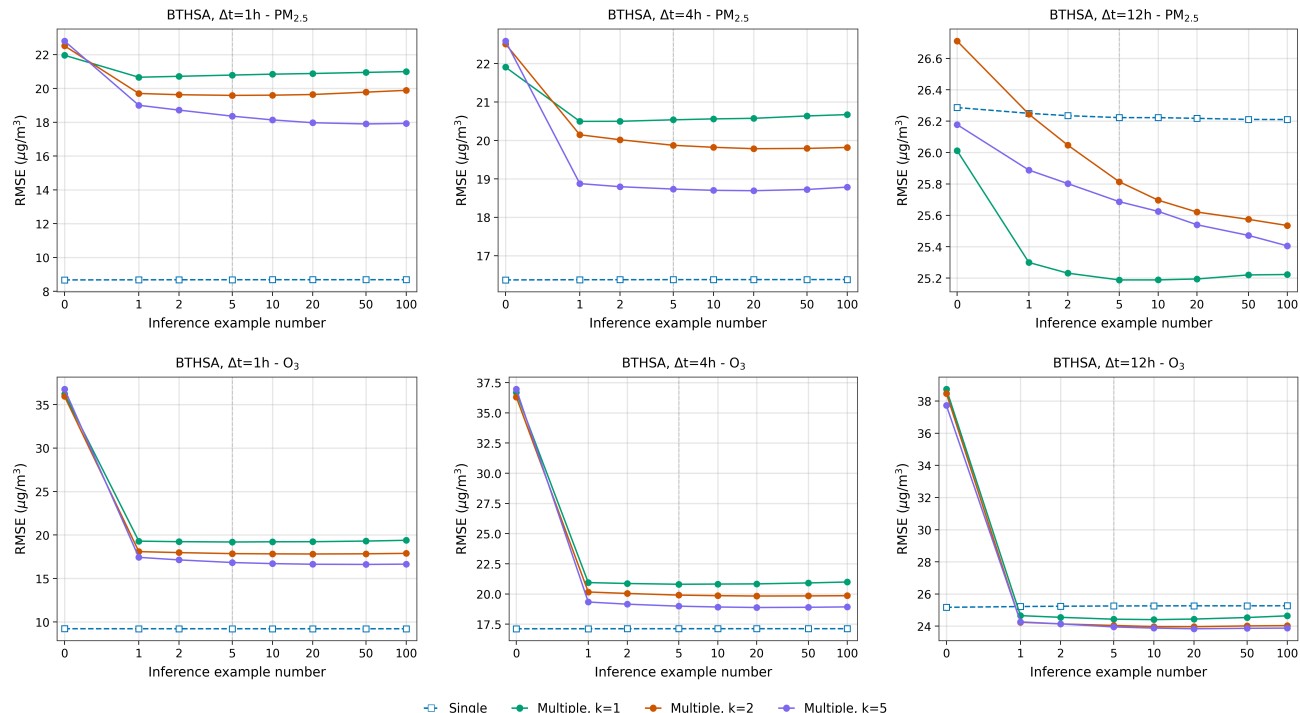

*Figure 5.* Example cardinality generalisation on BTHSA for simple to moderate operators. Top: PM$_{2.5}$. Bottom: O$_3$. Left to right: $\Delta t = 1, 4, 12$h. Classical single-operator learning achieves lower RMSE for simple operators ($\Delta t = 1, 4$h), while ICON with operator diversity outperforms the baseline at $\Delta t = 12$h given sufficient examples. All models are evaluated with up to 100 examples despite training with at most 5.

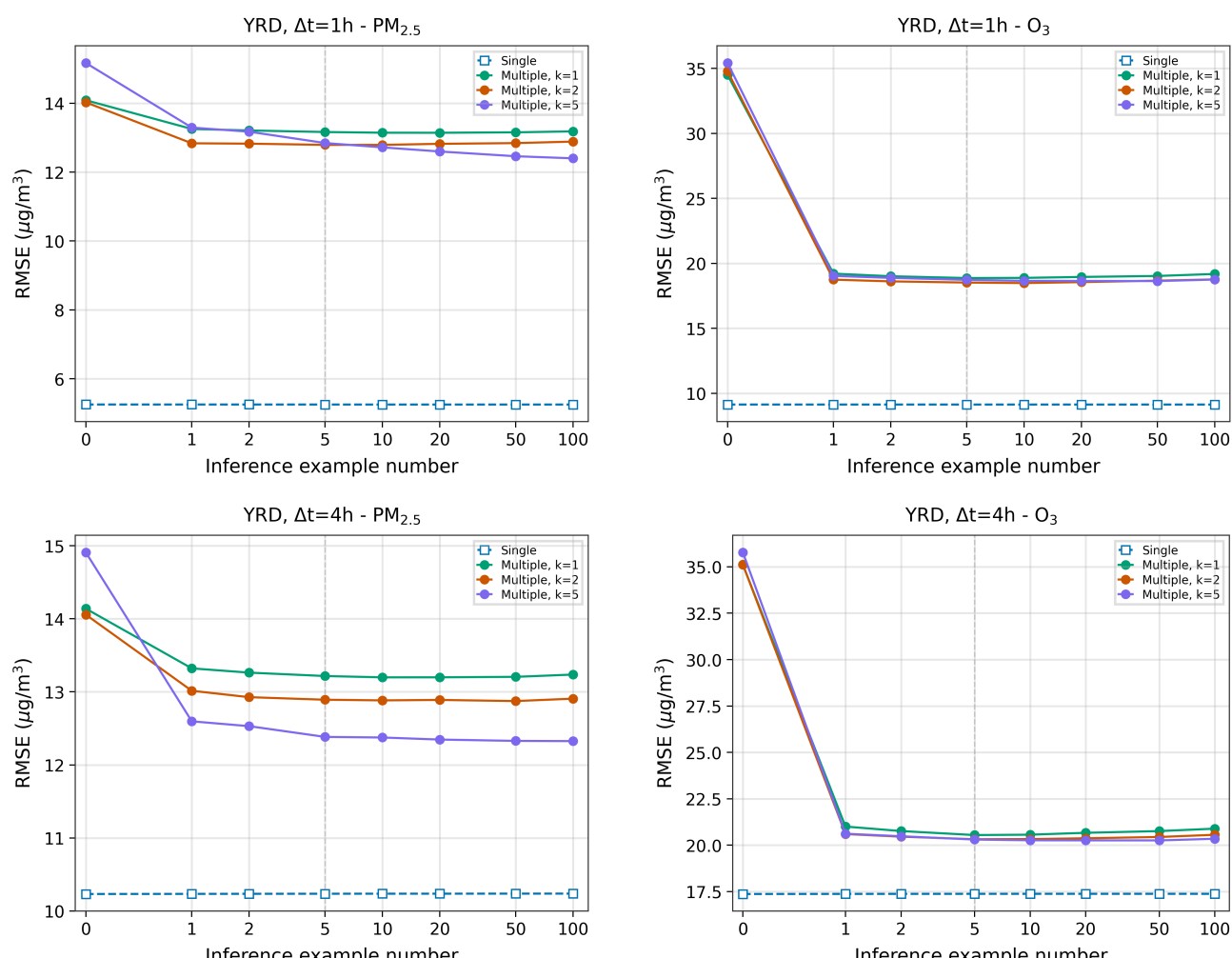

*Figure 6.* Example cardinality generalisation on YRD for simple operators. Left: PM$_{2.5}$. Right: O$_3$. Top: $\Delta t = 1$h. Bottom: $\Delta t = 4$h. Classical single-operator learning achieves lower RMSE for these simple operators.

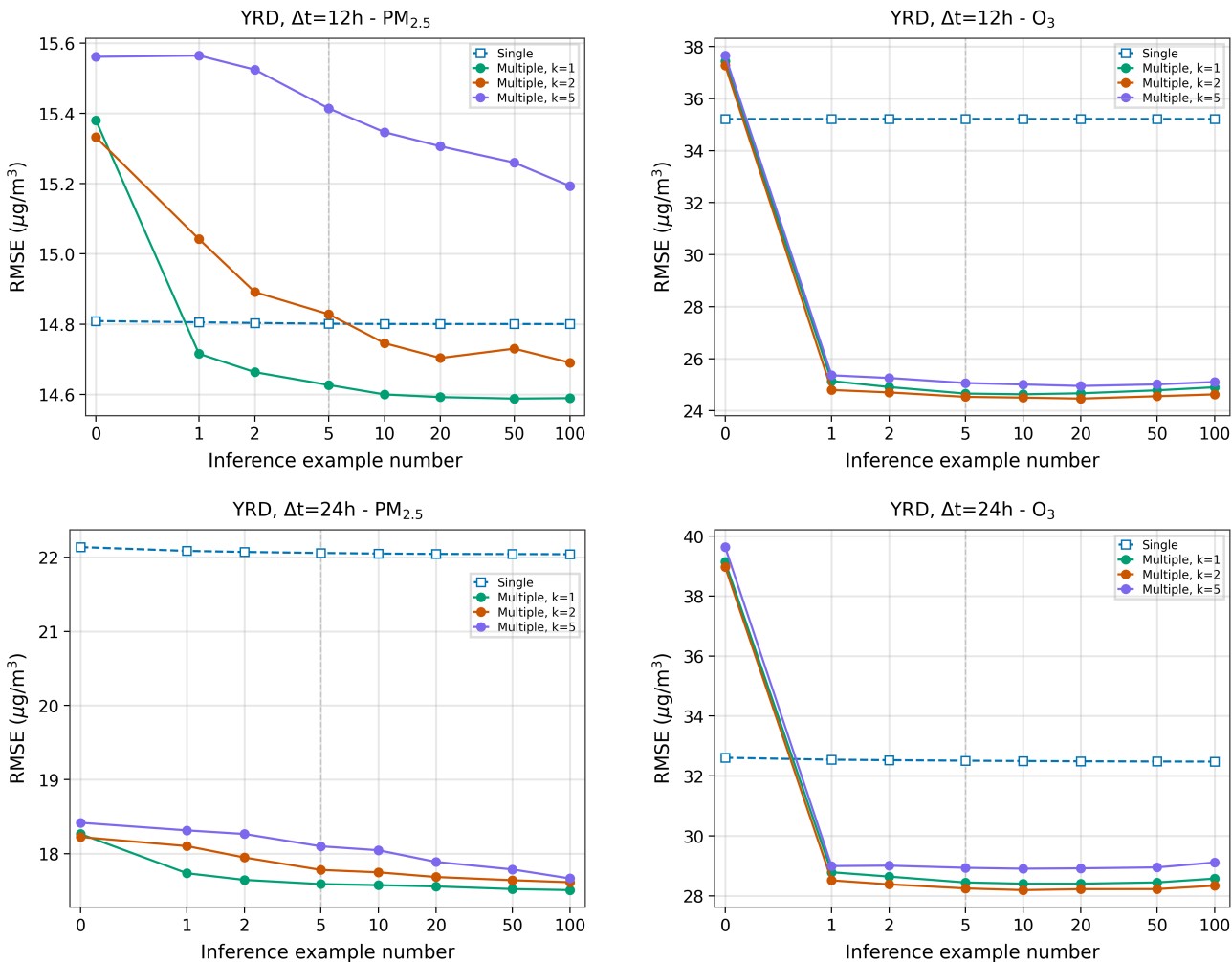

*Figure 7.* Example cardinality generalisation on YRD for complex operators. Left: PM$_{2.5}$. Right: O$_3$. Top: $\Delta t = 12$h. Bottom: $\Delta t = 24$h. ICON with operator diversity surpasses the baseline with sufficient examples.

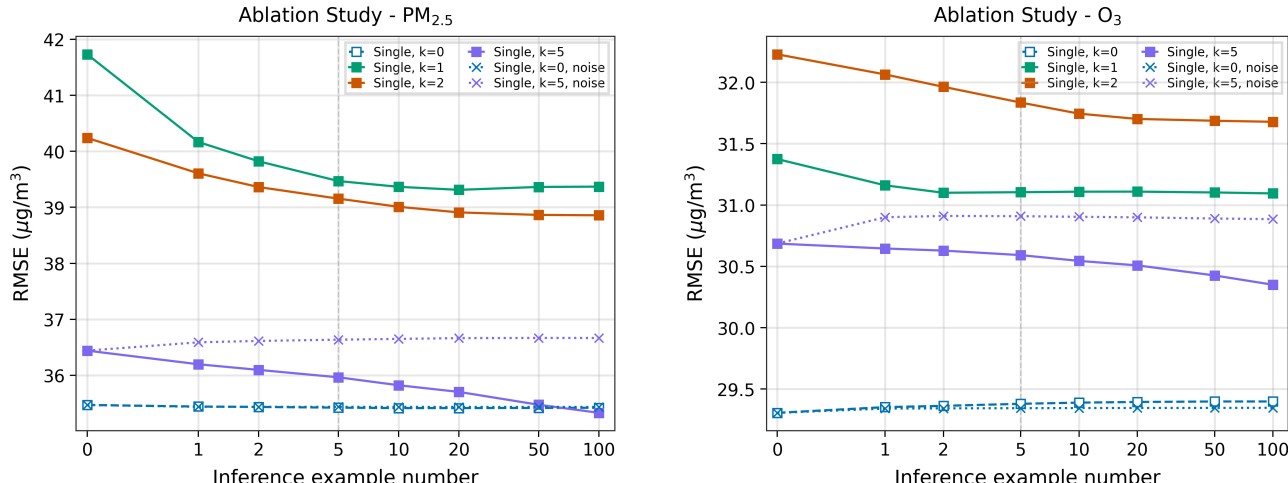

*Figure 8.* Single-operator setting ($\Delta t = 24$, on BTHSA). Left: PM$_{2.5}$. Right: O$_3$. Models trained without contextual examples show no difference between noise and quality examples. The model trained with $k = 5$ approaches and slightly surpasses the baseline for PM$_{2.5}$ at 100 quality examples. When given noise, models trained with examples show clear degradation, confirming sensitivity to example content.

