# OpenReview forum: "Graph In-Context Operator Networks for Generalizable Spatiotemporal Prediction"
_ICML.cc/2026/Workshop/FMSD — FMSD @ ICML 2026 Poster_

### Official Review · Reviewer_jFY5 · 2026-05-13

**Rating:** 6
**Confidence:** 4

**Review:**

## Summary

This paper introduces graph in-context operator network, GICON, which improves upon the in context operator network, which aims to solve operator learning problem via in-context learning, without retraining.
By introducing the message passing layer in the architecture, the proposed methodology can be applied to irregular geometry (non-grid, or non-rectangular domain) and expected to have resolution or geometry generalisation.
Authors also introduce the carefully designed positional encoding that allows the network to relate two items in sequence that correspond to query-key pairs, and discriminate which is key or value.


## Strength

Using message passing architecture in neural operator learning is standard approach, and the paper concretely combines the recent in-context operator learning architecture to graph neural operators.
Authors also introduce a positional encoding mechanism tailored for the in-context operator setting, which handles history-prediction pair’s relation, also with graph level information for the attention.
Suggested methodology outperforms the single operator setting even with single inference example, and improves with increasing number of in-context examples, generalises to other unseen operators, and also generalises over different graph structure.
Overall, the suggested methodology shows in-context operator learning can not only benefit on regular grid setting, but also on irregular setting, which is closer to the practical prediction problems.


## Areas for Improvement

It is unclear how well this methodology is performing. While I haven't worked with such air quality dataset,  the data from my local meteorological administration shows 0~60 PM2.5 index, therefore, RMSE of 34 seems a bit too high, and questionable if this is a reasonable baseline or performance. One way to show it is performing well would be providing standard deviation of each index to support how small the RMSE is.

It seems the standard choice in in-context-operator-learning is utilising the causal attention, which is different to TabPFN or TabICL, where they allow contexts to attend to each other.
While this loses the training efficiency, it also gains the ability to generalise over size of data.
The distinct loss curve, especially plateau in O3 prediction might be the result of failing length generalisation, and this may help.

Authors also described that the model trained with $k=5$ achieves the best extrapolation.
However, $k=5$ was the maximum training context size that authors employed.
Is it expected that increasing $k$ will improve the extrapolation ability more? Or $k=5$ is the sweet spot?
I think showing benefit of in-context operator learninf is already helpful in workshop paper, but if authors are considering future direction, identifying whether some kind of scaling law appears for training context size appears or dimishing return occurs, would be crucial.

The use of specific positional encoding, I believe, should be tested against the ablation study.
In [Lin et al. 2025], without specially designed positional encoding, ALiBi, even NoPE shows strong in-context-learning performance, with moderate length generalisation ability.
Both graph level attention bias and key / value specific token generalisation would require justification.

[Lin et al. 2025] How Do Position Encodings Affect Length Generalization? Case Studies On In-Context Function Learning, AAAI 2025.

## Detailed Comments

In Figure 2, left and center plots are models evaluated in BTHSA, but the right plot is model evaluated in YRD, trained on BTHSA. To maintain the consistency and possibly comparison between plots, it would be better to fix the evaluation dataset to BTHSA, and include models trained on YRD.

It is helpful to include the training dynamics, the overfitting behaviour is a bit concerning.
It would not be a problem if authors included sufficient hyperparameter tuning for both baseline and suggested methodology, but overfitting can be interpreted that the baseline is not sufficiently tuned.

## Justification of Score

The paper is definitely interesting to this community.
Incorporating in-context learning on graph structured data is one of the challenges, and while the tested problem is limited to operator learning, the suggested graph-level attention bias can be a possible direction.
The suggested methodology and the evaluation methods are also rigor, and authors’ comparison with single-operator learning problems shows the benefit of utilising in-context learning.

There is one concern regarding the performance benefit of this architecture, whether the performance gain here is nontrivial, and whether the network actually learned something.

---

### Official Review · Reviewer_KrwH · 2026-05-17
**Good architectural innovations but requires additional ablation studies and benchmarking experiments.**

**Rating:** 7
**Confidence:** 3

**Review:**

Strengths:
- This work address a gap in the ICON literature wherein they present a controlled comparison under matched training steps.
- The paper presents two architectural innovations graph message passing and example aware encoding.
- Both the above innovations are well motivated and cleanly presented

Weakness
- Single operator models are able to see each simple around 24 times more often as compared to multi operator models. The authors are suggested to explain the confusion between matching training steps and matching training information. Rectify the controlled comparison if necessary.
- Spatial context is frozen before the in-context learning step within each layer. It is possible that leads to a loss of expressivity compared to joint attention( even though joint attention is more expensive).
- The evaluation is limited to air quality with only two regions and target variables. The claims require a more comprehensive evaluation. Though this might be acceptable for a workshop paper, the authors are encouraged to expand the study.
- Similarly, the paper only compares against classical single operator learning. The authors are encouraged to compare against GNN based autoregressive models and spatiotemporal transformers. Also, only RMSE values are reported. Prior works in this air quality forecasting use multiple metrics.

Despite the fairness concerns and lack of error bars, the paper does present a novel architecture with graph message passing with in-context operator learning. Thus, it is a good workshop paper.